# Development of Scaffolds with Adjusted Stiffness for Mimicking Disease-Related Alterations of Liver Rigidity

**DOI:** 10.3390/jfb11010017

**Published:** 2020-03-14

**Authors:** Marc Ruoß, Silas Rebholz, Marina Weimer, Carl Grom-Baumgarten, Kiriaki Athanasopulu, Ralf Kemkemer, Hanno Käß, Sabrina Ehnert, Andreas K. Nussler

**Affiliations:** 1Department of Traumatology, Siegfried Weller Institute, Eberhard Karls University, 72076 Tübingen, Germany; silasrebholz@gmail.com (S.R.); marina-weimer@live.de (M.W.); carlgrom-baumgarten@web.de (C.G.-B.); sabrina.ehnert@gmail.com (S.E.); andreas.nuessler@gmail.com (A.K.N.); 2Faculty of Applied Chemistry, Reutlingen University, 72762 Reutlingen, Germany; kiriaki.athanasopulu@reutlingen-university.de (K.A.); ralf.kemkemer@gmail.com (R.K.); 3Faculty of Basic Science, University of Applied Sciences Esslingen, 73728 Esslingen am Neckar, Germany; hanno.kaess@hs-esslingen.de

**Keywords:** scaffold culture, stiffness, in vitro model, pre-coating, Arg-Gly-Asp (RGD)-peptides, cell attachment

## Abstract

Drug-induced liver toxicity is one of the most common reasons for the failure of drugs in clinical trials and frequent withdrawal from the market. Reasons for such failures include the low predictive power of in vivo studies, that is mainly caused by metabolic differences between humans and animals, and intraspecific variances. In addition to factors such as age and genetic background, changes in drug metabolism can also be caused by disease-related changes in the liver. Such metabolic changes have also been observed in clinical settings, for example, in association with a change in liver stiffness, a major characteristic of an altered fibrotic liver. For mimicking these changes in an in vitro model, this study aimed to develop scaffolds that represent the rigidity of healthy and fibrotic liver tissue. We observed that liver cells plated on scaffolds representing the stiffness of healthy livers showed a higher metabolic activity compared to cells plated on stiffer scaffolds. Additionally, we detected a positive effect of a scaffold pre-coated with fetal calf serum (FCS)-containing media. This pre-incubation resulted in increased cell adherence during cell seeding onto the scaffolds. In summary, we developed a scaffold-based 3D model that mimics liver stiffness-dependent changes in drug metabolism that may more easily predict drug interaction in diseased livers.

## 1. Introduction

Today the testing of new drugs is mainly performed in animals due to a lack of predictive in vitro models [1]. However, animal experiments have several limitations in predicting liver toxicity of new substances, due to differences between the drug-metabolizing enzymes in humans and animals [2]. Therefore, new models that are able to foresee the in vivo situation more accurately are needed. Freshly isolated human hepatocytes are the gold standard for such models since their metabolic profile is comparable to the in vivo environment [3,4]. However, the use of human liver cells for the testing of new substances is limited since human hepatocytes are scarcely available and suffer frequently from the loss of metabolic function during long-term cultivation [5,6]. In recent years continuously available liver cell lines were established as an alternative to human hepatocytes to overcome these limitations [7]. 

Furthermore, numerous attempts have been made to improve cultivation conditions to extend the hepatocytes metabolic activity and to improve the metabolic activity of these cell lines [4,8,9,10]. Various studies showed that mimicking the in vivo environment improves the metabolic capacity of primary liver cells and liver cell lines [4,8,11,12]. In addition to the interaction with other cells that can be provided in a co-culture setup [13], it is also important that the hepatocytes interact with the surrounding matrix [14]. One reason for the rapid loss of the metabolic properties of primary hepatocytes in conventional 2D cultures is the lack of adequate interaction with the surrounding matrix [12]. In recent years, various 3D cultivation methods have been developed, including scaffold culture systems [12,15]. These 3D culture systems can help to maintain the metabolic activity of cells over an extended period of time [4,16]. The metabolic activity is sustained because the cultivation of these cells on scaffolds ensures that cell adherence is not limited to the plane of the cell culture plate but can take place three-dimensionally. Moreover, the porous surface of the scaffold improves the cell nutrients’ supply. Although several studies have shown that substrate stiffness greatly influences the functioning of cells, this parameter has been neglected so far in the development of scaffolds for the cultivation of hepatocytes. For instance, one study showed that mesenchymal stem cells could be differentiated into different directions only by using different rigidities [17]. The influence of substrate stiffness on the functioning of hepatocytes is known [18,19]. In line with these publications, it has been shown that rat hepatocytes cultured onto a soft 2D matrix (2 kPa) resulted in a more differentiated and functional hepatocyte phenotype than those cultured directly onto a stiffer surface (55 kPa) or cell culture plastic. Moreover, on day 7, cytochrome P450 (CYP) activity was 2.7 times higher in hepatocytes cultured on the softer matrix than those cultured on normal cell culture plastic [11]. Interestingly, in vivo, the metabolic activity of the hepatocytes is also influenced by liver stiffness. Theile et al. observed a significant difference between the activity of drug metabolic enzymes in patients with a liver stiffness lower than 8 kPa compared to patients with a liver stiffness larger than 8 kPa; this result indicates that the hepatic stiffness directly affects the metabolism of drugs in vivo [20]. 

Since such metabolic changes caused by different liver rigidities between healthy subjects and patients with liver fibrosis or cirrhosis may also influence the efficiency and/or toxicity of drugs, these adverse effects should be considered either during drug development or drug prescription to patients with liver diseases. Therefore, the first aim of this study was the development of scaffolds that mimic the stiffness of healthy and fibrotic livers. In addition to stiffness, it is well established that the extracellular matrix (ECM) influences the properties of liver cells [4,21]. In order to provide cells with an optimal environment for attachment and during cultivation on the scaffold, we used different proteinaceous solutions and various incubation intervals to pre-coat scaffolds. We decided to use the cell line HepG2 for establishing and testing our 3D liver scaffolds since cell line is often used as an alternative to primary human hepatocytes in drug metabolism studies [22].

## 2. Materials and Methods 

### 2.1. Production of the Different pHEMA Based Scaffolds

For a successful culture of hepatic cells, it is necessary to generate a matrix whose properties are as similar as possible to those of the in vivo environment. The ideal matrix allows good cell attachment, therefore it must have pores of a sufficiently large diameter, high porosity, and permeability to allow cells to penetrate and adhere [16,23]. Further, a sufficient supply of nutrients must be guaranteed [24]. To achieve this goal, we used poly-(2-hydroxyethyl methacrylate) (pHEMA)/bisacrylamide (BAA)-based cryogels. To increase cell adherence, collagen (self-made rat-tail collagen), concentration 3.5 g/L prepared as described [25] and gelatin solution (30% cold water fish gelatin in ddH_2_O) were added to the scaffolds. Several different cryogel compositions were tested. The various concentrations of cryogel components that were tested during the development of the scaffolds are shown in Appendix A. The four that most closely met the above-mentioned criteria regarding the scaffold architecture, which could in addition be reliably reproduced, were selected for further testing. The exact ingredient compositions of the four different scaffold prototypes are listed in Table 1.

The cryogels were prepared as shown in Figure 1. In brief, the gels were prepared by thoroughly mixing collagen, ddH_2_O, and gelatin and then incubating on ice for at least 30 min. Then pHEMA and BAA were added and the solution was thoroughly mixed again. Polymerization was started by adding tetramethylethylenediamine (TEMED), aminoperoxodisulfate (APS), and glutaraldehyde. Then, 10 mL of solution, containing all substances listed in Table 1, were immediately pipetted into several 2 mL syringes, which were placed in a freezer at −18 °C for a minimum of 16 h to allow the polymerization of the scaffold components to form the cryogel. After polymerization, the frozen scaffolds were cut into 3 mm-thick slices. The scaffolds were sterilized in 70% ethanol under agitation for 12 h and then soaked three times in PBS for at least 2 h, to remove the ethanol and non-polymerized scaffold components. Preliminary experiments showed that pre-incubation of the scaffolds for 72 h with medium containing 10% fetal calf serum (FCS) increased cell adherence significantly.

### 2.2. Scaffold Characterization

#### 2.2.1. Pore Size

Analyses of pore size and pore structure were performed as previously described [16]. Briefly, the scaffolds were stained using a sulforhodamine B (SRB) solution (0.08% SRB in 1% acetic acid). The scaffolds were washed three times with 1% acetic acid solution to remove unbound SRB. The pore structure of the cryogels was visualized by using red fluorescence to reveal bound SRB. An EVOS fluorescence microscope (Life Technologies, Darmstadt, Germany) was used to obtain fluorescent images. The size and shape of the pores were determined using ImageJ software, version 1.5 (National Institutes of Health, Bethesda, MD, USA).

#### 2.2.2. Porosity

The porosity of the various scaffolds was measured using a modified version of the protocol published by Fan et al. [26]. Briefly, scaffolds, with a height of about 3 mm and a diameter of 1 cm that were fully saturated with PBS for at least 30 min were weighed (m_1_) and placed in a cell strainer. After centrifugation for 10 min at 4200× *g* the scaffolds were weighed again (m_2_). The porosity was calculated using the following formula:$$\mathrm{Porosity}\;\left(\%\right)=\frac{\left(m1-m2\right)\;}{\;\left(\rho w\times V\right)}\times 100\%\;V=scaffold\;volume,\;\rho w=density\;of\;PBS.\;$$

#### 2.2.3. Scaffold Permeability

The permeability of the scaffolds was determined by measuring the liquid diffusion rate [27]. Briefly, the scaffolds were transferred to a well plate to which enough SRB solution (0.08% in 1% acetic acid) was added to ensure that the scaffolds were surrounded, but not covered. The diffusion of the red-colored SRB solution into the scaffold was measured over 30 min. Microscopic images were captured every 5 min using a reflected light microscope (Bresser, E122002, Rhede, Germany). The distance from the scaffold border to the dye front was measured and evaluated using ImageJ. At each time-point, the distance was asssed at 20 sites per scaffold. The diffusion rate per minute was calculated. Representative images which illustrate the procedure are shown in Appendix A.

#### 2.2.4. Water Uptake Rate and Swelling Ratio

The water uptake rate and swelling ratio of the scaffold prototypes, with a height of about 3 mm and a diameter of 1 cm, were measured as previously described [28] and calculated using the following equations:(1)$$\mathrm{Swelling}\;\mathrm{ratio}\;\left(\%\right)=\frac{\left(\mathrm{scaffold}\;\mathrm{wet}\;\mathrm{weight}\;\left(g\right)-\;\mathrm{scaffold}\;\mathrm{dry}\;\mathrm{weight}\;\left(g\right)\right)\;}{\mathrm{scaffold}\;\mathrm{dry}\;\mathrm{weight}\;\left(g\right)\;}\times 100$$
(2)$$\mathrm{Water}\;\mathrm{uptake}\;\left(\%\right)=\frac{\left(\mathrm{scaffold}\;\mathrm{wet}\;\mathrm{weight}\;\left(g\right)-\;\mathrm{scaffold}\;\mathrm{dry}\;\mathrm{weight}\;\left(g\right)\right)}{\mathrm{scaffold}\;\mathrm{wet}\;\mathrm{weight}\;\left(g\right)}\times 100$$

The dry and wet weights (in grams) of the scaffolds were measured with an analytical balance as described above.

#### 2.2.5. Matrix Stiffness

A microscale mechanical testing system (Microsquisher, CellScale, Waterloo, Canada) was used to measure the mechanical properties of the various scaffolds [29]. The scaffolds were stored in PBS before measurement to prevent dehydration. PBS soaked scaffolds were used for compression testing. Cylindrical samples with a height of about 3 mm and a diameter of 2 mm were cut out of the different scaffold prototypes. All samples were mechanically compressed with a calibrated tungsten microbeam (diameter 0.558 mm) with a 3 × 3 mm square compression plate at the end. The samples were uniaxially compressed by 10% with force and displacement. Details are given in Table 2

These force-displacement data were used to calculate the modulus of elasticity (E) by dividing the nominal stress value (σ) by the maximal nominal strain (ε) of the samples.

### 2.3. Scanning Electron Microscope (SEM) Images

#### 2.3.1. Sample Preparation for the SEM Images

The samples of healthy and cirrhotic livers were used for the SEM images from patients that had undergone tumor resection surgery [6]. The samples were frozen in liquid nitrogen immediately after removal. Informed, written consent was obtained from patients in accordance with the ethical guidelines of the Ethics Commission of the medical faculty of the University of Tübingen, Tübingen, Germany (project number: 298_2012BO1). SEM images of the scaffolds and the liver tissue were obtained after lyophilization.

#### 2.3.2. Preparation of the SEM Images 

SEM investigations were performed using a JEOL JSM-7200 FLV setup with a field-emission source (JEOL, Freising, Germany). Prior to imaging, the samples were sputtered with gold. The thickness of the sputter layer was about 20 nm. First, a series of images for each sample was taken to provide an overview. The illustrations presented in this report are from characteristic areas selected from these images. The size of the structures shown can be determined from the scale bars included in the images.

### 2.4. Culture of HepG2 Cells and Seeding of the Cells on the Scaffold

The HepG2 cells were cultured as previously described [30]; briefly, the cells were cultivated in a Dulbecco’s Modified Eagle’s Medium (DMEM) high glucose medium containing 10% FCS, 100 U/mL penicillin, and 100 µg/mL streptomycin. All cell culture reagents were purchased from Sigma Aldrich (St. Louis, MI, USA). The cells were cultured at 37 °C in a humidified atmosphere with 5% CO_2_. HepG2 cells from passages two to ten after thawing were used for the experiments as previously described [31]. Before plating, the cells for the experiments were washed with PBS and detached from the cell culture flask using trypsin/EDTA (0.5 g/L trypsin and 0.2 g/L EDTA). Cell detachment was checked with the light microscope, and then trypsin digestion was stopped using FCS-containing culture medium. The cells were centrifuged at 600× *g* for 10 min, then the supernatant was removed, and the cells were resuspended using fresh culture medium. Using a Neubauer chamber, the cells were counted and seeded on the scaffolds in the desired concentration using the “drop-on” seeding method [16,30]. Scaffolds were placed in a 24-well plate, as much of the medium as possible that had been used for the pre-incubation was aspirated and 40 μL of the cell suspension were dispensed onto the central area on the top of the scaffolds. Then, 700 μL of fresh culture medium were added to the cells after 4 h. This amount of media was required to cover the scaffold completely. 

### 2.5. Measurement of Mitochondrial Activity with Resazurin

To measure Resazurin conversion, the scaffolds were washed once with PBS and then incubated with a 0.0025% Resazurin solution (in DMEM medium) for 1 h at 37 °C. The fluorescence of the resorufin thus produced was measured at 544 nm/590-10 nm using the Omega Plate Reader (BMG LABTECH, Ortenberg, Germany) [16].

### 2.6. Staining of the Cells with Calcein-AM and Hoechst

The cells cultured on the scaffolds were stained with Calcein-AM (final concentration 2 µM) and Hoechst 33342 (final concentration 2 µg/mL) to enable fluorescence microscopy images to be captured. Hoechst dye was used to stain double-stranded DNA; it allows cell nuclei to be detected in the fluorescence microscope channel DAPI (357/447 nm). Calcein-AM was used to stain living cells and was detected in the green fluorescent protein (GFP) channel (470/525 nm). A mixture of both dyes diluted in PBS was added to the cell-seeded scaffolds and incubated for 30 min at 37 °C, protected from light. Then, scaffolds were washed at least three times with PBS. Microscopy of stained cells was performed using the EVOS FL fluorescence microscope (Life Technologies, Darmstadt, Germany).

### 2.7. Effect of Scaffold Pre-Incubation

#### 2.7.1. Increasing Cell Attachment by Pre-Incubation of Scaffolds

Several solutions were tested to improve the cell adherence by pre-incubation of the scaffolds. We tested Arg-Gly-Asp (RGD)-rich proteinaceous solutions, such as gelatin and human serum, as well as culture media with and without FCS. For this experiment, the scaffolds were pre-incubated for at least 7 days with the RGD-containing solutions. As a control condition scaffolds were incubated in PBS for the same period. The substances used and their concentrations are shown in Table 3. The cells were seeded on the scaffolds in a density of 2 × 10^5^ cells/scaffold as described before, and the conversion of Resazurin was measured after 24 h.

#### 2.7.2. Length of Pre-incubation of Scaffolds with Culture Medium

For this experiment, scaffolds were prepared as described before. Three scaffolds per experiment were transferred to FCS-containing culture medium at 14, 10, 7, 3, 2, and 1 day(s) before cells were plated on the scaffolds. The remaining scaffolds continued to be incubated in PBS, and 8 × 10^4^ HepG2 cells were plated per scaffold (day 0). A reduction of the cell numbers relative to the previous experiment was required because in this experiment we were not only investigating the cell attachment but also the viability of the cells over 5 days cultured on the scaffold. Resazurin conversion was measured 24 h as well as on day 5 after seeding. Additionally, living cells were stained for microscopy using Calcein-AM on day 5.

### 2.8. Metabolic Tests of the Cells on the Scaffolds

We measured the activity of different phase I/II enzymes as well as the ability of cells to detoxify ammonia as indicators of hepatic function. Cells were seeded onto scaffolds at a concentration of 2 × 10^5^ cells/scaffold as described above. Since previous studies revealed that the addition of insulin and hydrocortisone to the culture medium increases the activity of CYP enzymes in liver cell lines [32], we supplemented our medium with 1 mM human insulin and 0.8 µg/mL hydrocortisone 24 h after seeding [32]. Then, cells were incubated on scaffolds for a total time of 72 h since recent studies revealed maximum metabolic changes at this time point [32]. The subsequent metabolic tests were performed as follows.

#### 2.8.1. Urea Measurement

The quantification of urea was carried out using a 3D adaptation [16] of a protocol published by Seeliger et al. [33]. The scaffolds were washed with PBS 72 h after seeding, then the scaffolds were incubated using an additive-free medium for 24 h in the presence of 5 mM ammonium chloride. Then, 80 µL of the supernatant were mixed with 60 µL of O-phthalaldehyde solution (1.5 mM O-phthalaldehyde, 4 mM Brij-35, 0.75 M H_2_SO_4_) and 60 µL of NED reagent (2.3 mM N-(1 Naphthyl) ethylenediamine dihydrochloride, 80 mM boric acid, and 4 mM Brij-35, 2.25 M H_2_SO_4_), and incubated for 1 h at 37 °C. The absorbance was measured at 460 nm (Omega Plate Reader) and compared to a urea standard curve (0–100 µg/mL) on the same plate.

#### 2.8.2. Measurement of Phase I/II Activities

The activity of phase I/II enzymes was measured by fluorescence-based methods, as described by Ehnert et al. [34]. The activity of phase I enzymes CYP 1A2, CYP 3A4, and CYP 2C9, and phase II enzymes, uridine diphosphate glucuronosyltransferase (UGT) and glutathione S-transferase (GST), was measured since these enzymes are essential in the metabolism of several drugs [4,35]. The substrates, the phase II inhibitors, and the products used, as well as the corresponding wavelengths, are summarized in Table 4. Before measuring, the scaffolds were washed once with PBS and then incubated for 30 min in plain medium-containing substrates and inhibitors as indicated. To exclude the interference of scaffold components with the measurement, scaffolds without cells were used as a background control.

### 2.9. Statistical Analysis

Non-parametric Mann–Whitney U tests were used to assess differences between two groups. Group comparisons involving more than two groups were carried out with the non-parametric Kruskal–Wallis H test, followed by Dunn’s multiple comparison test, implemented in GraphPad Prism 5.00 Software (GraphPad Software, San Diego, CA, USA). Data are represented as means ± SEM from at least three independent experiments (N ≥ 3). All statistical comparisons were two-sided as this was an exploratory data analysis and significance is denoted as follows: *p* < 0.05 (*), *p* < 0.01 (**), and *p* < 0.001 (***).

## 3. Results

### 3.1. Characterization of the Natural ECM of Healthy and Cirrhotic Liver Tissue

To develop scaffolds corresponding to the healthy and fibrotic liver, it is necessary to characterize the respective in vivo environments and then develop an in vitro model representing these characteristics. Therefore, we captured SEM images of healthy and cirrhotic liver tissue samples (Figure 2A,B). The images show that there are differences between the structure of the ECM of the healthy and cirrhotic liver. The ECM of the healthy liver is, as described before [6], an open-pored, thin-walled structure (Figure 2A), while the cirrhotic tissue has thicker cell walls and slightly larger pores (Figure 2B).

### 3.2. Testing of Different Scaffolds for the Cultivation of Liver Cells 

To mimic the in vivo environment as closely as possible, we carried out preliminary tests on various different pHEMA/BAA-based scaffold prototypes, created by systematically varying single scaffold components. Table 5 and Figure 3 summarize the characteristics of the scaffolds that have a porous structure, allow cell attachment, and match the criteria for pore size and stiffness best. Therefore, they were selected for further investigation. Details of the used scaffolds components and their quantities are given in Table 1. The pore size of the different scaffold prototypes differed significantly between various scaffolds. Scaffold prototype 1 exhibited the largest pores, with a mean pore diameter of 115 ± 29 μm, whereas the pores of scaffold prototype 4 had a mean diameter of only 61 ± 41 µm. Scaffold prototype 1 also had the highest water uptake capacity and highest swelling ratio, followed by scaffold prototype 3.

Scaffold prototype 3, with a stiffness of 2.9 ± 1.3 kPa, showed rigidities similar to a healthy liver [36,37], whereas scaffold prototypes 1 and 4 showed the stiffness characteristics of a fibrotic or cirrhotic liver [36,37] (Figure 3A). As shown in Figure 3B, the adherence of the cells to the different scaffolds was tested by seeding different amounts of HepG2 cells (between 5 × 10^4^ and 2 × 10^5^ cells) on the scaffolds. Attachment of living cells was determined 24 h after seeding by measuring the conversion of Resazurin. Additionally, as shown in Figure 3C, cell attachment to the scaffolds was visualized using Calcein-AM and Hoechst 33342 staining, which causes fluoresce of living cells. Both the microscopic images and the Resazurin measurements showed that cells attached to all the scaffolds tested. Overall cells seeded on scaffold prototype 3 showed the highest conversion of Resazurin. Although all the scaffolds presented here could be used to cultivate HepG2 cells, scaffold prototypes 1 and 3 were deemed the most promising for our purposes due to their pore size, Resazurin turnover, and especially, stiffness. The stiffness of healthy livers can be mimicked with scaffold prototype 3 (healthy liver scaffold), while fibrotic/cirrhotic livers are represented by scaffold prototype 1 (fibrotic liver scaffold).

The selected scaffolds were further characterized in terms of porosity and permeability. The healthy liver scaffold had significantly higher porosity than the fibrotic liver scaffold (Figure 4A), but the two scaffolds had a similar permeability (Figure 4B). In addition, the SEM images revealed that both scaffolds showed a porous structure, with slightly larger pores in the fibrotic liver scaffold. However, both scaffolds had significantly larger pores than the corresponding human liver tissue (Figure 4C). 

### 3.3. Effect of Scaffold Pre-incubation on Cell Adherence and Their Viability During the Culture

We investigated whether pre-incubation of the scaffolds in culture media containing various RGD-containing substances can increase the adherence of the cells to the scaffolds. As can be seen in Figure 5, pre-incubation had a positive effect on cell adherence in both tested scaffolds. However, the results clearly showed that this effect was much more pronounced in healthy liver scaffolds. There, pre-incubation increased cell attachment by as much as ten times, but in the case of the fibrotic liver scaffold, only a 1.4-fold increase in cell adherence was achieved. The results indicate that due to pre-incubation with FCS-containing medium, the cell adherence to both scaffold types could be increased, which is why this type of pre-incubation was used for both scaffolds later in the study.

In order to provide cells with an ideal environment for attachment on scaffolds and during culture over several days, we investigated the optimal time period for scaffold pre-incubation. Since previous results showed that pre-incubation had a more pronounced effect in ‘healthy’ liver scaffolds, this scaffold type was used to determine the ideal pre-incubation time. To determine the most suitable time period, the scaffolds were pre-incubated over periods of 1, 2, 3, 7, 10, and 14 day(s) using medium containing 10% FCS and 8 × 10^4^ HepG2 cells were seeded on the scaffolds. Resazurin conversion was measured 24 h and five days after seeding. On day five living cells on the scaffold were additionally stained with Calcein-AM. As shown in Figure 6, the length of the pre-incubation had a huge influence on cell attachment properties. In addition, the number of living cells cultured on the scaffolds was significantly influenced after five days by the length of the pre-incubation period. A pre-incubation period of 24–72 h resulted even after a culture period of five days only in a low number of viable cells viable on the scaffold (Figure 6C). In addition, we observed a low level of Resazurin turnover after 24 h (Figure 6A) as well as after five days in culture (Figure 6B). However, using a pre-incubation period of seven days or more, many more living cells attached to the scaffold (Figure 6C). This result was accompanied by a significantly higher Resazurin turnover after 24 h (Figure 6A) as well as after five days (Figure 6B) in culture.

As described above, the coating of scaffolds, especially the healthy liver scaffold, significantly increased cell adherence. SEM images were captured and analyzed to determine whether this change was accompanied by a modification of the scaffold surface. To do this, both scaffold types were pre-incubated for seven days in FCS-containing medium or PBS (control condition) and then freeze-dried to avoid damaging of the surface structure. As seen in Figure 7, the surface structures of the two types of scaffold differed significantly. The healthy liver scaffold had a rougher surface on which crystal-like constituents were superimposed, whereas the fibrotic liver scaffold appeared rather smooth, with only small crystals superimposed on the surface. Pre-incubation with the FCS-containing medium did not significantly alter the surface of the fibrotic liver scaffold, but in contrast, resulted in significant changes in the healthy liver scaffold. Large deposits of agglomerates were observed on the surface of healthy liver scaffolds incubated in FCS-containing medium.

### 3.4. Evaluation of the Functionality of Hepatic Cells Plated on the Scaffolds

After characterizing the scaffolds, we investigated if stiffness variations had an impact on the metabolic activity of the seeded cells. As depicted from Figure 8, the cultivation of cells on ‘healthy’ liver scaffolds led to increased metabolic activity of CYP 3A4, CYP 2C9, and UGT compared to cells seeded onto the fibrotic liver scaffold. In contrast, the activity of CYP 1A2, and to a certain extent, GST was higher in cells seeded onto fibrotic liver scaffolds compared to cells cultured onto ‘healthy’ liver scaffolds. There was no statistically significant difference in urea production between cells cultured either on healthy or fibrotic scaffolds. 

## 4. Discussion

Disease-specific changes in the liver, that can be found in liver fibrosis and cirrhosis, alter the metabolism of drugs [38]. This not only necessitates individualized therapies but also the use of models that can represent these individual differences during the development and testing of drugs. This can be achieved in animal research through the use of certain disease models [39], but it is not yet possible for in vitro research. Therefore, the aim of this study was the development of such an in vitro model that mimics the in vivo situation of a “healthy” and a fibrotic altered liver. Scaffolds are suitable to imitate the in vivo environment since the cells are cultivated in a three-dimensional environment. It is also possible to imitate the interaction of cells with the ECM. In addition, it is possible to generate scaffolds that correspond in their stiffness to so-called “healthy” livers and a more fibrotic altered liver. This is an important issue since ‘healthy’ livers have a stiffness of approx. 6 kPa, which is much lower compared to the stiffness of cell culture plastics of approx. 100,000 kPa [20,40]. As our results show, it is possible to mimic the rigidity of the human liver and the stiffness of a fibrotic liver using pHEMA/BAA-based scaffolds. The scaffolds we tested were based on the same substances but differed in their gelatin content. By increasing the gelatin concentration, more gelatin molecules were covalently cross-linked by glutaraldehyde to form more stable amide bonds, thus increasing the stiffness of cryogels [41]. This is of particular interest since gelatin is an irreversibly hydrolyzed collagen and therefor has similar binding sites for cells [42,43]. Moreover, an increase in collagen fibers is an important aspect to develop various models of liver fibrosis and cirrhosis [44,45]. Thus, in addition to rigidity, a high ECM deposition can be mimicked by the fibrotic liver scaffolds [46]. It is likely that the higher gelatin concentration of the fibrotic liver scaffold is responsible for the fact that when scaffolds were not subjected to pre-incubation, with medium-containing FCS, cell adherence was significantly higher for the fibrotic scaffold than the healthy liver scaffold [27]. Pre-incubation with FCS-containing medium led to the deposition of agglomerates on the surface of ‘healthy’ liver scaffolds, but not of fibrotic-like liver scaffold surfaces. The accumulation of this agglomerates in the healthy liver scaffold and changes the surface structure of scaffolds; it seems it therefore may be responsible for increased cell attachment [47]. Pre-incubation experiments lined-out that culture medium alone was able to increase cell adherence. Whether the amino acids content in the medium and/or other medium components was responsible for this positive attachment effect of cells onto the scaffold surface cannot be determined from the SEM images. However, as suggested by Amirikia et al., pre-incubation of scaffolds results in an increase in protein adsorption to the scaffold surface, thus reducing scaffold roughness and water contact angle, and therefore leading to an increased attachment of cells [47]. The same explanation could also account for the positive effect of pre-incubation which we found in our “healthy” liver scaffold.

When comparing the physical characteristics of both scaffold types (healthy and fibrotic-like), it was noticeable that the fibrotic-like liver scaffold had larger pores than the ‘healthy’ liver scaffold. Both scaffolds had significantly larger pores than the corresponding human liver tissues. Having a larger pore size relative to the in vivo analog could be desirable because in the in vitro model nutrients are not supplied to the cells via the bloodstream. This is in line with previous research that suggested that larger pore size may result in faster cell growth and increased diffusion [16,23,48]. In addition to this, both scaffolds have similar permeability but show different porosities. However, as can be seen from the SEM image, under dry conditions, the fibrotic-like liver scaffold also has an open pore structure. The low porosity value of this scaffold is probably due to water binding to the included gelatin. This could lead to a swelling of the matrix and in turn, reduce the volume of its pores. The lower porosity of the fibrotic liver also seems to be affected by the increased ECM deposition [46], as shown in the SEM images. This scaffold characteristic tends to make the model more similar to the in vivo environment. Since the scaffolds have similar permeability, one would not expect the difference in porosity to affect nutrient supply. As described above, the scaffolds were designed to represent liver stiffness of the healthy as well as the fibrotic human liver [20]. Unfortunately, we were not able to determine the stiffness of human liver samples using the same method used for the scaffolds. Therefore, the measured rigidities were compared to literature data determined by other methods [36,37].

Our activity measurements indicate that the cells plated onto softer ‘healthy’ liver scaffolds had higher levels of activity of phase I enzymes CYP 3A4 and CYP 2C9 and phase II enzyme, UGT 2B7; whereas the activity of the phase I enzyme CYP 1A2 and the phase II enzyme GST was higher in cells cultured on fibrotic-like liver scaffolds. The differences between the scaffold types with respect to phase II metabolism is a very interesting finding as it is already known that in liver cirrhosis, respectively steatosis, UGT 2B7 activity is reduced whereas GST activity is induced [49,50]. This result is remarkable since a change in phase II metabolism can lead to the formation of alternative metabolites with different pharmacological and toxicological potentials [51]. When evaluating these results, it must also be taken into account that HepG2 cells were used in this study, which have some of the functions of human hepatocytes, but differ significantly from them, especially in their metabolic activity [31,52]. Further studies are required to rule out that drug and/or substrate compounds might be absorbed to the scaffold surface and therefore deliver false-positive or false-negative results [53]. Although the differences between the two scaffolds concerning stiffness, pore size, and surface structure capture the in vivo differences well, the metabolic differences observed in the clinic cannot be attributed solely to the difference in stiffness. It might be useful to develop an in vitro model system that represents other aspects of liver fibrosis pathology, such as the change in ECM composition, oxygen tension, or inflammation [6,54,55]. Future 3D liver models should include cells like hepatic stellate and/or Kupffer cells to better mimic liver fibrosis [46].

## 5. Conclusions

Within this study, we developed scaffolds capturing the stiffness of a healthy and fibrotic-like liver. The coating of healthy liver scaffolds led to a change in the scaffold surface, which was accompanied by a significant increase in cell adherence. In addition, a correlation between the length of the pre-incubation period and the increase in cell adherence was observed. Since there were stiffness-related differences in the metabolic activity of the cells on the two scaffold types, they can be considered as an excellent model for studying how fibrotic-like liver alterations affect the metabolism of drugs.

## Figures and Tables

**Figure 1 jfb-11-00017-f001:**
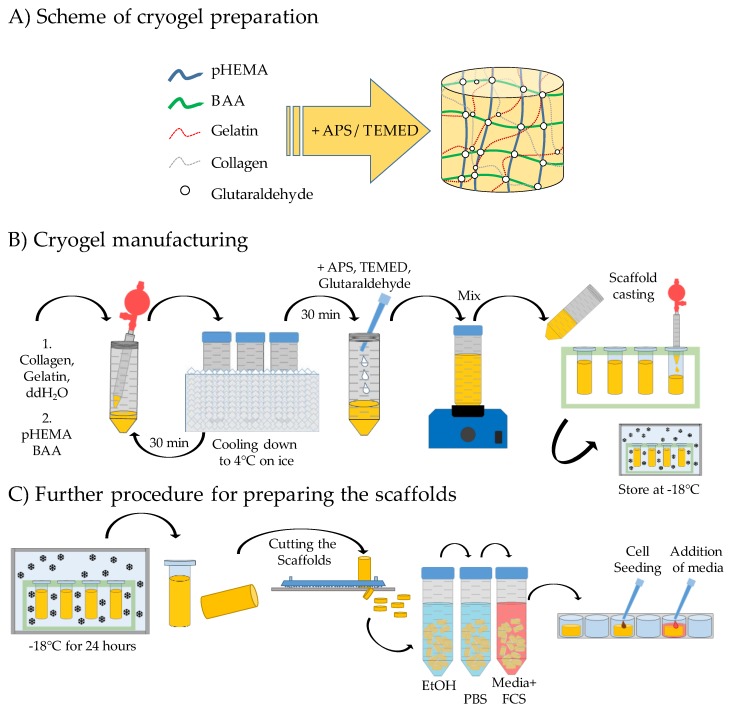
Schematic overview of scaffold production. (**A**) Formation of a scaffold matrix through the cross-linking of the scaffold components. (**B**) Mixture of the individual scaffold components and pouring of the scaffolds. (**C**) Further procedure, cutting the scaffolds, preparation of the scaffolds prior to cell cultivation, and seeding of the cells.

**Figure 2 jfb-11-00017-f002:**
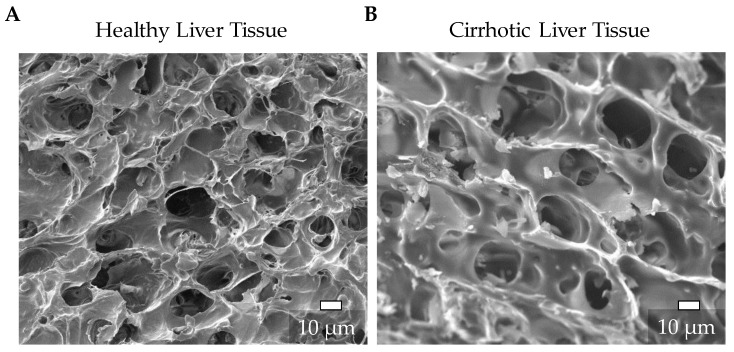
Representative SEM images of the extracellular matrix (ECM) structure of healthy (**A**) and cirrhotic liver (**B**) tissue (scale bar 10 µm).

**Figure 3 jfb-11-00017-f003:**
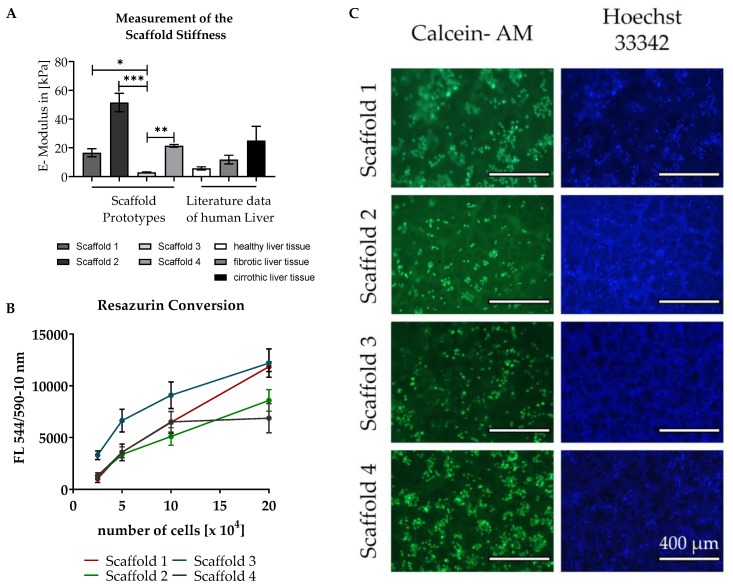
Characterization of the scaffold prototypes. (**A**) Scaffold stiffness was measured in N = 3, n = 3, since we were not able to measure the stiffness of liver tissue until now, the values were compared with published data [36] created by Fibroscan measurements. Bars represent mean ± SEM; *p* < 0.05 (*), *p* < 0.01 (**), *p* < 0.001 (***) as indicated. (**B**,**C**) Attachment of cells to the four different scaffold prototypes was tested. HepG2 cells were plated in various concentrations on four different scaffold prototypes and cell attachment was evaluated by measuring Resazurin conversion and Hoechst 33342/Calcein-AM staining 24 h after seeding. The Resazurin conversion data are represented as averages of three independent experiments (each *n* = 3). Lines represent mean ± SEM. Hoechst 33342 and Calcein-AM staining is shown in representative fluorescence microscopy images of 1 × 10^5^ cells/scaffold.

**Figure 4 jfb-11-00017-f004:**
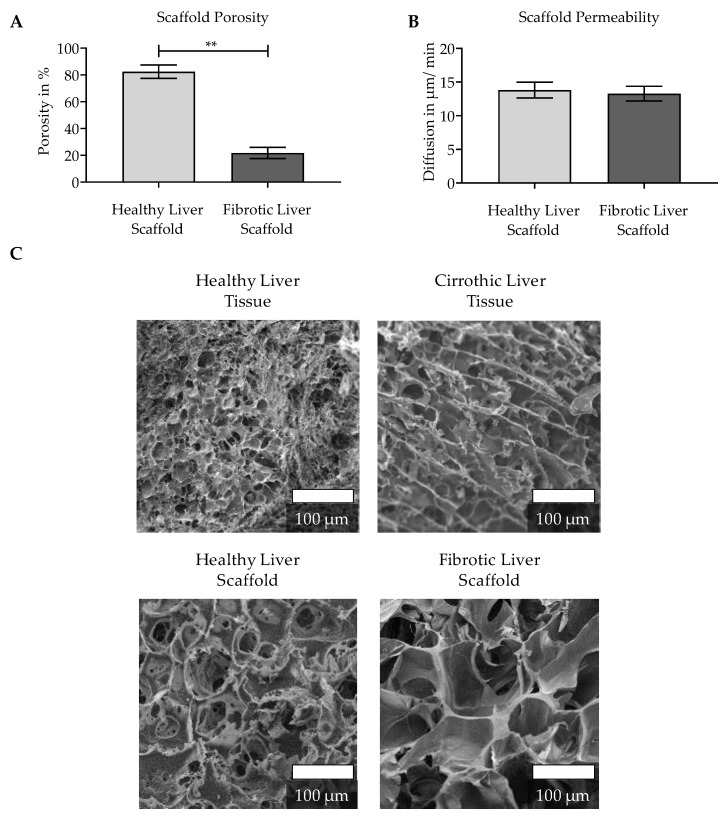
Physical characterization of the pHEMA/BAA-based 3D scaffolds: (**A**) Determination of the scaffold porosity (*n* = 5), three scaffolds were pooled and measured together. (**B**) Permeability was measured in a total of 9 scaffolds from three independent scaffold manufacturing days. Bars represent mean ± SEM; *p* < 0.01 (**) as indicated. (**C**) SEM images of healthy liver and cirrhotic altered liver tissue and both selected scaffolds (scale bar 100 µm).

**Figure 5 jfb-11-00017-f005:**
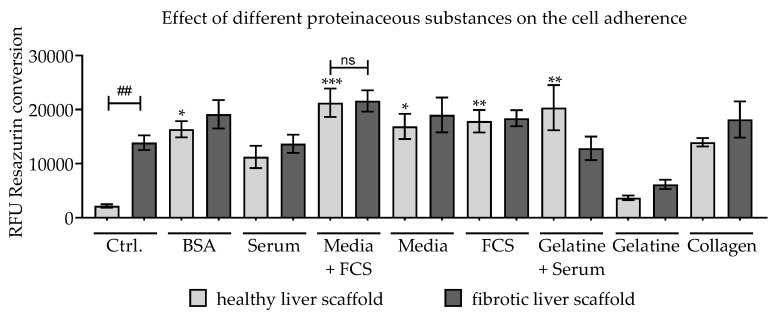
Effect of pre-incubation with various proteinaceous substances. Scaffolds were pre-incubated for at least 7 days with the Arg-Gly-Asp (RGD)-containing solutions. Resazurin conversion was measured 24 h after plating 2 × 10^5^ HepG2 cells/scaffold on pre-coated scaffolds; scaffolds pre-incubated in PBS were used as a control; N = 3, n = 2. Bars represent mean ± SEM. The significance of differences between pre-incubation with PBS (control) and proteinaceous substances is denoted as follows: **p* < 0.05, ***p* < 0.01, ****p* < 0.001. The difference between the not pre-incubated healthy and fibrotic liver scaffold is indicated as ##*p* < 0.01. The difference between the media and FCS pre-incubated healthy and fibrotic liver scaffolds are indicated as not significant (ns).

**Figure 6 jfb-11-00017-f006:**
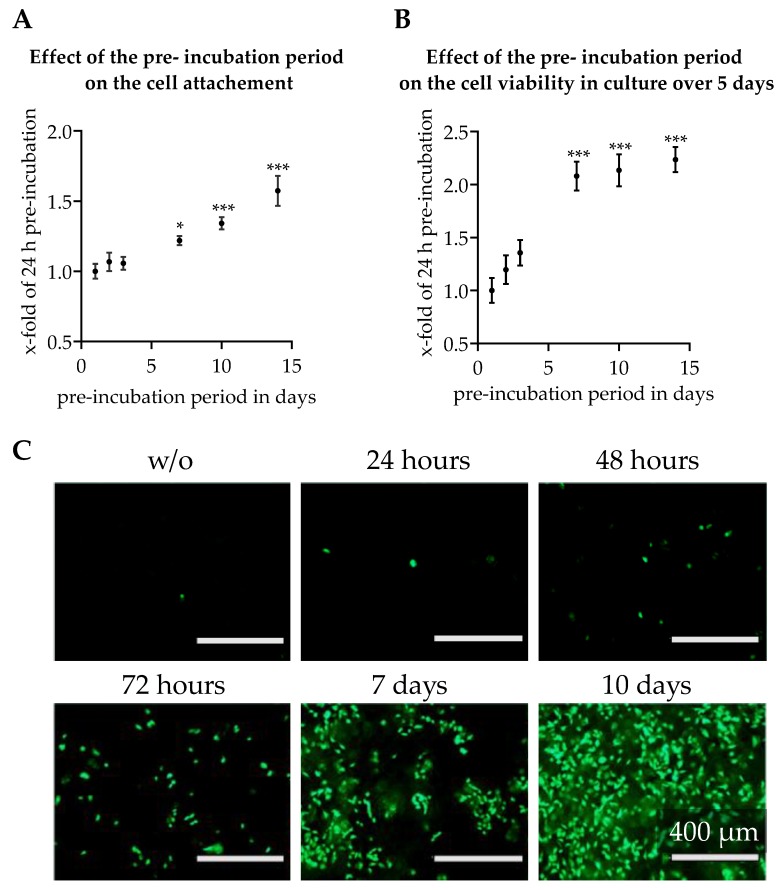
Effect of the length of pre-incubation on the attachment and the maintenance of HepG2 cells plated on the healthy liver scaffold. The scaffolds were pre-incubated with FCS-containing media between 24 h and 14 days. HepG2 cells were seeded on scaffolds. The conversion rate of Resazurin was measured 24 h (**A**) and five days (**B**) after seeding. The graphs show mean ± SEM for N = 3, n = 3; **p* < 0.05, ****p* < 0.001 for comparison with a pre-incubation period of 24 h as indicated. Calcein-AM staining (**C**) after a culture period of five days; representative images of HepG2 cells plated on scaffolds using different pre-incubation periods.

**Figure 7 jfb-11-00017-f007:**
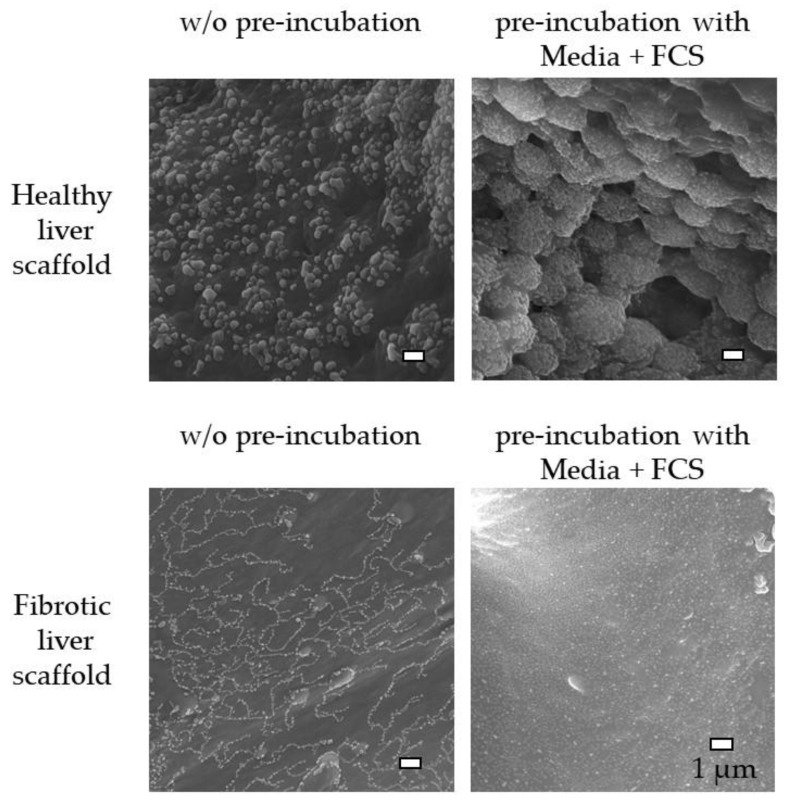
Effect of pre-incubation on the surface of the scaffolds. Representative SEM images of the surface of healthy and fibrotic liver scaffolds with and without (w/o) 7-day pre-incubation in FCS-containing medium (scale bar 1 µm).

**Figure 8 jfb-11-00017-f008:**
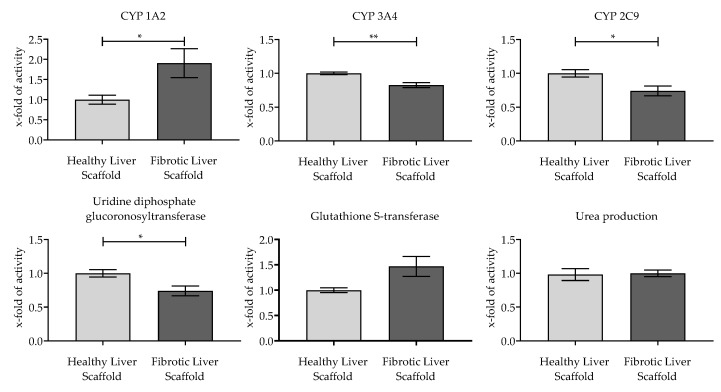
Metabolic activity of HepG2 cells plated on the two scaffolds. The activity of phase I enzymes CYP 1A2, CYP 3A4, and CYP 2C9, and phase II enzymes, UGT and GST, was measured. The detoxification of ammonia was quantified by measuring the urea production. Values are shown as multiples of the activity of cells plated on the healthy liver scaffold, N = 3, n = 3; bars represent mean ± SEM; **p* < 0.05, ***p* < 0.01 as indicated.

**Table 1 jfb-11-00017-t001:** Quantities of the substances used for cryogel formation.

Substance Concentration	Scaffold 1	Scaffold 2	Scaffold 3	Scaffold 4
ddH_2_O	5.78 mL	3.78 mL	7.11 mL	5.03 mL
pHEMA 98%	500 µL	500 µL	1 mL	1 mL
BAA 2%	250 µL	250 µL	170 µL	10 µL
Gelatin 30%	2 mL	4 mL	250 µL	2 mL
Collagen 3.5 g/L	1 mL			
TEMED	20 µL			
APS 10%	200 µL			
Glutaraldehyde 25%	250 µL			

poly-(2-hydroxyethyl methacrylate), 2-hydroxyethyl methacrylate; BAA, bisacrylamide; TEMED, tetramethylethylenediamine; APS, aminoperoxodisulfate.

**Table 2 jfb-11-00017-t002:** Measurement parameters for mechanical properties for pHEMA scaffolds.

Magnitude	Loading Rate	Hold	Recovery	Rest	Repeats
10.0%	10%/100 s	2 s	10 s	0 s	1

**Table 3 jfb-11-00017-t003:** Substances used in the pre-incubation experiment.

Substance	Concentration	Note
DMEM	100%	Without additives
DMEM, FCS, P/S	10% FCS, 1% P/S	10,000 units penicillin and 10 mg streptomycin/mL
Collagen	0.14 g/L in PBS	This concentration is usually used for plate coating [25]
Bovine serum albumin	5% in PBS	-
Coldwater fish gelatin	30% in ddH2O	-
Human serum	100%	-
FCS	100%	-
Gelatin, FCS	50% FCS, 15% Gelatin, ddH_2_O	-

DMEM, Dulbecco’s Modified Eagle’s Medium; FCS, fetal calf serum; P/S, Penicillin/ Streptomycin.

**Table 4 jfb-11-00017-t004:** Conditions used for phase I/II measurements.

Enzyme	Substrate	_C_ final in µM	Measured Product	Phase II Inhibitors	Measured Wavelength
CYP 1A2	7-Ethoxycoumarin	25	7-Hydroxycoumarin	1.5 mM Salicylamid, 2 mM Probenecid	355/460 nm
CYP 3A4	7-Benzyloxy-4 (trifluoromethyl) coumarin	5	7-Hydroxy-4 (trifluoromethyl) coumarin	1.5 mM Salicylamid, 2 mM Probenecid	355/520 nm
CYP 2C9	Dibenzylfluorescein	5	Fluorescein	10 µM Dicumarol	485/520 nm
UGT 2B7	4-Methylumbelliferon	6.25	4-Methylumbelliferon	-	355/460 nm
GST	Monochlorobimane	50	Monochlorobimane- glutathione conjugate	-	355/460 nm

**Table 5 jfb-11-00017-t005:** Characteristics of the four scaffold prototypes.

Measured Parameter	Scaffold 1	Scaffold 2	Scaffold 3	Scaffold 4
Pore diameter (µm)	115 ± 29	64 ± 13	85 ± 41	61 ± 41
Water uptake (%)	90.4 ± 1.6	83.9 ± 1.9	85.3 ± 4.4	83.4 ± 2.0
Swelling ratio (%)	965 ± 159	528 ± 72	641 ± 207	511 ± 74

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
