# Peer review of "Development of Scaffolds with Adjusted Stiffness for Mimicking Disease-Related Alterations of Liver Rigidity"

_jfb, 2020, doi:10.3390/jfb11010017_

Round 1

Reviewer 1 Report

The manuscript „Development of scaffolds with adjusted stiffness for mimicking disease-
related alterations of the liver rigidity” showed new developed scaffolds that
mimic the rigidity of healthy and fibrotic liver tissue. The MS is suitable for publication after minor revisions: 

1. The authors decided to use the cell line HepG2 for the experiments, which is a hepatocellular carcinoma cell line. Although it is an alternative to primary human hepatocytes, this is a cancer cell line. HepG2 in the scafford prototype 3 would represent the healthy liver tissue and cells. Are there any differences in the adhesion or metabolism between HepG2 and normal cells? Please at least comment on this.
2. There is no information about the passage number, however, it influences the amount of integrin receptors on the cells. (Integrins are transmembrane receptors that facilitate cell-extracellular matrix (ECM) adhesion). Please give more information about this.
3. In the statistical analysis, the data are represented as means ± SEM. Why do the authors used SEM instead of SD? Please at least comment on this.

Author Response

Dear Editor, dear Reviewers,
We like to thank the editor and reviewers for giving us the opportunity to re-revise our manuscript. We appreciated your comments and have addressed all issues regarding your recommendations and edited the manuscript accordingly. The detailed answers to the reviewer comments are summarized below. All changes performed in the manuscript are highlighted. We would like to resubmit our revised manuscript to Journal of Functional Biomaterials and hope that the revised manuscript meets now the high standards of this journal.
All authors have approved the final submitted manuscript and no conflict of interest exists between the authors
Yours sincerely,
Prof. Dr. Andreas K. Nüssler

Reviewer 2 Report

The authors present a detailed exploration of scaffold design to facilitate liver tissue engineering, by leveraging the understanding of the physical/mechanical environment of the healthy and fibrotic liver. This work merits publication in this journal after addressing these minor concerns:

Introduction

  • Break up the paragraph into 2-3 paragraphs. I suggest starting a new paragrph at line 42, "Therefore, numerous attempts.." and and new paragraph starting at line 67, "Since such metabolic..."

Discussion

  • start new paragraph at line 452, "Our activity measurements..."

Author Response

(The authors gave the same response as above.)
